# Perfluorooctanesulfonate Can Cause Negative Bias in Creatinine Measurement in Hemodialysis Patients Using Polysulfone Dialysis Membranes

**DOI:** 10.3390/membranes12080778

**Published:** 2022-08-13

**Authors:** Wen-Sheng Liu, Chien-Hung Lin, Szu-Yuan Li, Chih-Ching Lin, Tsung-Yun Liu, Ann Charis Tan, Han-Hsing Tsou, Hsiang-Lin Chan, Yen-Ting Lai

**Affiliations:** 1Division of Nephrology, Department of Medicine, Taipei City Hospital Zhongxing Branch, Taipei 103, Taiwan; 2School of Medicine, National Yang Ming Chiao Tung University, Hsinchu 300, Taiwan; 3College of Science and Engineering, Fu Jen Catholic University, New Taipei City 242, Taiwan; 4Institute of Food Safety and Health Risk Assessment, National Yang Ming Chiao Tung University, Hsinchu 300, Taiwan; 5Department of Special Education, University of Taipei, Taipei 100, Taiwan; 6Department of Pediatrics, Taipei Veterans General Hospital, Taipei 112, Taiwan; 7Division of Nephrology, Department of Medicine, Taipei Veterans General Hospital, Taipei 112, Taiwan; 8Kim Forest Enterprise Co., Ltd., New Taipei City 221, Taiwan; 9Department of Child Psychiatry, Chang Gung Memorial Hospital and University, Taoyuan 333, Taiwan; 10College of Medicine, National Taiwan University, Taipei 100, Taiwan; 11Department of Physical Medicine and Rehabilitation, National Taiwan University Hospital Hsin-Chu Branch, Hsinchu 300, Taiwan

**Keywords:** creatinine, Jaffe method, enzymatic method, isotope dilution liquid chromatography mass spectrometry, hemodialysis, perfluorooctanoic acid, perfluorooctanesulfonate

## Abstract

Serum creatinine is an important clinical marker for renal clearance. However, two conventional methods (Jaffe and enzymatic) are prone to interferences with organic compounds as compared to the standard method (isotope dilution–liquid chromatography–mass spectrometry) and can cause a significant negative bias. Perfluorooctanesulfonate (PFOS) and perfluorooctanoic acid (PFOA) are two common perfluorochemicals (PFCs) that can easily be accumulated in humans. We aimed to verify whether this bias is the result of an accumulation of PFCs. The serum creatinine values of 124 hemodialysis patients were analyzed using the three methods. We also aimed to evaluate which biochemical parameters will influence the difference between the conventional methods and the standard method. We found that a significant underestimation occurred when using the conventional methods. Albumin is an independent factor associated with negative bias, but it loses this correlation after dialysis, likely due to the removal of protein-bound uremic toxins. PFOS can cause negative bias when using the enzymatic method. Furthermore, this linear correlation is more significant in patients who used polysulfone-based dialysis membranes, possibly due to the better clearance of other uremic toxins. The serum creatinine of uremic patients can be significantly underestimated when using conventional methods. PFCs, as well the type of dialysis membrane being used, can be influencing factors.

## 1. Introduction

Inulin clearance is considered to be the most accurate determination of renal function [1,2]. However, it is difficult to perform the procedure on a regular basis. Therefore, serum creatinine and its clearance is generally used as a convenient marker to evaluate renal function [3]. Serum creatinine is the most commonly used method to determine renal function in the clinical setting, but serum creatinine values can be influenced not only by the method of measurement but also by other confounding factors [4]. There are two methods available clinically: the more popular Jaffe method [5] and the enzymatic method. There is also the gold standard method, which is the isotope dilution–liquid chromatography–mass spectrometry (IDLCMS) method [6,7]. 

The Jaffe method utilizes oxidation to form a chromophore in serum creatinine measurement in the laboratory. However, it can be easily confounded with chromogens such as nitromethane, as well as glucose and protein with positive bias [8,9,10,11]. On the other hand, the enzymatic method is less affected by glucose. It is useful to check creatinine levels in high-glucose-containing peritoneal dialysate. However, the enzymatic method is still influenced by certain organic compounds [6]. Among the three methods, IDLCMS is the most accurate, but it is expensive, which limits its application [12,13,14,15]. In a healthy population, these three methods show a good linear correlation.

Uremic patients whose serum creatinine levels are higher than the normal population suffer from renal failure and are subjected to an accumulation of toxic waste [16,17]. The linear correlation of the three different methods in uremic patients was tested in our previous study in 2012, and it was found that albumin acts as a negative bias in uremic serum instead of a positive bias in normal serum. Positive bias means that the measured value is higher than the real value and negative bias means the measured value is lower [18]. In that study, we proposed a theory that the reason albumin acts as a negative bias may be related to protein-bound uremic toxins. We hypothesized that these toxins which cause negative bias may accumulate in uremic patients and be bounded by albumin, but failed to find proof of this over several years. In another previous study, in 2018, we found that perfluorochemicals (PFCs) are higher in uremic patients and are mainly carried by albumin [19]. The most two common PFCs are perfluorooctanoic acid (PFOA) and perfluorooctanesulfonate (PFOS), which are also persistent organic pollutants. These compounds with long half-lives are difficult to eliminate from the human body and may cause eventual endocrine disturbance [20,21], Furthermore, not all dialysis membranes can provide adequate clearance for PFCs. Results from a previous study showed that polysulfone dialysis membranes provide better clearance for PFCs [21].

Because there is no study at the moment that proves that protein-bound toxins may interfere with serum creatinine measurement in uremic patients, the aim of this study is to determine whether serum PFCs may cause errors in the Jaffe and enzymatic methods in uremic patients. We also aim to investigate how different dialysis membrane properties and the clearance of PFCs may influence the serum creatinine measurement error in the Jaffe and enzymatic methods. 

## 2. Materials and Methods

### 2.1. Inclusion and Exclusion Criteria

We included patients aged 18 to 90 who had been undergoing 4 h of routine hemodialysis three times per week for more than three months. Blood samples before and after dialysis were collected at the beginning of the month from a teaching hospital in northern Taiwan.

Patients who had undergone peritoneal dialysis and transplantation were excluded. Patients who had intravenous medications, such as lipid nutrition supplement, propofol, dopamine, methotrexate, vancomycin, furosemide, and cyclosporine were also excluded. Those who had blood loss with transfusion or admission were also excluded. A total of 124 patients were included.

### 2.2. Clinical and Biochemistry Data Collection

Demographic and clinical data such as gender, cause of renal failure (diabetes or chronic glomerulonephritis), age, and duration under hemodialysis were obtained from patients’ medical records. Laboratory parameters were gathered at the beginning of the month prior to hemodialysis. 

In this study, the liquid chromatography tandem mass spectrometry (LC-MS/MS) was used with an isotope dilution to quantify the serum creatinine, PFOA, and PFOS. The differences in creatinine, PFOA, and PFOS before and after dialysis were also evaluated. The hemogram auto-analyzer used was SYSMEX XE2100 (Sysmex, Kobe, Japan). The Jaffe and other biochemical parameters were determined by the ADVIA^®^ 1800 Chemistry System (Siemens Healthcare GmbH, Erlangen, Germany). The enzymatic method was determined by the Beckman AU640 (Beckman Coulter Inc., Brea, CA, USA) with amidohydrolase procedure. We used IDLCMS as the gold standard to evaluate the differences between the other two methods. The difference between Jaffe and IDLCMS is termed D-Jaffe and that of the enzymatic method and IDLCMS is termed D-enzymatic. 

First, we checked the creatinine values of the three methods. Then, we checked the association of D-Jaffe and D-enzymatic between clinical and biochemical factors by linear regression. Those significant factors were further analyzed with multiple linear regression to find the independent influencing factors for creatinine measurement. The post-dialysis values of independent influencing factors were analyzed again with D-Jaffe and D-enzymatic. Patients with different dialysis membranes were compared with their PFC levels and the association with D-Jaffe and D-enzymatic.

The procedures of measuring creatinine and PFCs is described in the supplement. The study was conducted in accordance with the Declaration of Helsinki and the protocol was approved by the Institutional Review Board of the hospital. 

### 2.3. Statistical Analysis

Continuous data were expressed as mean ± standard deviation. Statistical analysis was performed using IBM SPSS Statistics for Windows, Version 20.0 (IBM Corp., Armonk, NY, USA). The paired t-test was used to compare the difference before and after dialysis. The χ2 test was used for categorical variables and the t-test for continuous variables. Distributions of continuous variables in groups were expressed as mean ± SD. Multivariate forward logistic regression analysis was applied to identify the independent determinant factors. A p value of less than 0.05 was regarded as statistically significant.

## 3. Results

Table 1 and Figure 1 showed the value of serum creatinine measured in three methods and the difference between Jaffe and IDLCMS (D-Jaffe) and that of the enzymatic method and IDLCMS (D-enzymatic). The underestimation is about 10–15%. The creatinine values of the enzymatic method were also significantly lower than those of the Jaffe method due to the larger negative bias of D-enzymatic (Table 1). 

Table 2 showed that gender and DM as comorbidities did not affect the measurement error of serum creatinine. Continuous variables such as age, serum biochemical markers, and complete blood count index were also analyzed with linear regression to the D-Jaffe and D-enzymatic. (Table 2).

The factors which were significantly correlated with D-Jaffe and D-enzymatic were further analyzed with multiple linear regression in order to find the independent influencing factors (Table 3). The independent factors were serum albumin and serum Cr (both Jaffe and enzymatic method). As for PFCs, PFOA was not significantly removed by dialysis and did not affect the D-Jaffe and D-enzymatic values. On the other hand, PFOS only affected the enzymatic method. PFOS fulfilled the aforementioned requirement and was shown to cause negative bias in the enzymatic method pre-hemodialysis (Figure 2).

After dialysis, markers related to serum albumin, creatinine, and PFCs were analyzed in Table 4. Serum albumin lost its ability to associate with negative bias in both the D-Jaffe and D-enzymatic methods (Figure 3). There were lower PFOS levels post-dialysis and these did not affect serum creatinine values in the enzymatic method. However, serum creatinine was still strongly related with negative error for creatinine measurement.

Upon further analysis of dialysis membranes, PFOS causing negative bias in D-enzymatic were mainly seen in patients who used polysulfone-based dialysis membranes and this can be confirmed by the lower PFOS levels seen in patients using polysulfone-based dialysis membranes (Table 5). This may be related to the better clearance of uremic toxins and fewer confounding toxins. 

## 4. Discussion

We found significant negative measurement errors using the Jaffe method (D-Jaffe) and enzymatic method (D-enzymatic), as compared with the gold standard IDLCMS method. The error may cause approximately a 10–15% reduction compared to the real value, which is more significant when serum creatinine is high [18]. This finding explains why serum creatinine is not a sensitive marker to monitor renal function deterioration. The more toxins that are accumulated during advanced chronic kidney disease, the more they provide a negative bias, which offsets the expected increase during serum creatinine measurement [22].

Table 2 showed the factors that are significantly associated with D-Jaffe and D-enzymatic. After multiple linear regression, the two common independent factors related to the negative bias of D-Jaffe and D-enzymatic were serum albumin and the absolute value of Cr, as shown in Table 3. The post-dialysis association of independent factors related to D-Jaffe and D-enzymatic were shown in Table 4.

The fact that serum albumin lost its ability to cause negative bias post-dialysis was compatible with our hypothesis that at least one protein-bound uremic toxin may be removed by dialysis. Albumin is a negatively charged protein and a good carrier which binds with various cations and toxins, including PFCs [19] (Table 4 and Figure 3).

The other factor is creatinine, which is an indicator of glomerular filtration rate and renal function. Various toxins may accumulate in the bloodstream of uremic patients and some of them may not be dialyzable [19]. Hence, they may be still exist even after dialysis and produce negative bias on the serum creatinine value. Serum phosphorus was still significantly associated with the negative bias of serum creatinine measurement post-dialysis. This may be due to serum phosphorus also being associated with renal function decline, and only being able to be removed by half on regular hemodialysis [23]. On the other hand, serum potassium is easily removed by HD. Hence, serum potassium was not related with renal function post-dialysis.

As for PFCs, there is a higher concentration of PFOS than any other PFCs (such as PFOA) because of a longer history of usage [21]. Through multiple linear regression analysis, we found that PFOS was also an independent factor causing negative bias in the enzymatic method but not in the Jaffe method (Table 3). Upon further comparison of the differences between D-Jaffe and D-enzymatic, and analysis with linear regression, we found that PFOS was highly associated with this difference (*p* = 0.048, R = 0.226) and PFOS may be the reason why serum creatinine was lower in the enzymatic method as compared with the Jaffe method.

PFOS confirmed our previous hypothesis [18] that uremic toxins causing negative bias may be bound with albumin. After dialysis, the toxin is removed and albumin also simultaneously loses its association with the negative bias of serum creatinine. 

There are several factors that contribute to the results of this study. First, uremic toxins are composed of many materials, many of which are not yet fully understood. Second, the toxin concentrations may be too low to measure in an ordinary laboratory setting. IDLCMS was required to accurately measure PFCs [24]. The procedure for the measurement of serum creatinine and PFCs is described in the Appendix A; it requires appropriate equipment and well-trained staff. Third, the association between the dialysis membrane and clearance was also an important factor. There is a lower PFOS concentration in the serum of patients with polysulfone-based dialysis membranes compared with other materials (Table 5). This may be related to the better clearance of the polysulfone-based membrane for protein-bound toxins [24]. Polysulfone is more hydrophobic than other materials such as cellulose, and it has a better compatibility and clearance for toxins [20]. Therefore, the linear regression of PFOS and D-enzymatic were more obvious in this patient group due to fewer confounding toxins. As more studies showed the advantage of polysulfone-based membranes, it has gradually become the main dialysis membrane material in recent years [25]. In our current study, 75% of patients were using polysulfone-based dialysis membranes.

The limitation of the study is that we found only one toxin (PFOS) which causes a negative bias and is bound with albumin. However, it still confirmed our hypothesis, put forward in 2012, as to why albumin acts differently in the serum creatinine measurement of uremic patients. The strength of the study is that we used IDLCMS to evaluate the true value of serum creatinine, which is quite difficult and requires elaborate procedures. The concentrations of environmental toxins (PFOA and PFOS) were also too low to be measured in the ordinary laboratory setting. Furthermore, without the better clearance provided by polysulfone-based dialysis membranes, it will be impossible to find the linear association between PFOS and the negative bias measured in the enzymatic method because there are too many other confounding toxins found in the serum. More toxins and membrane materials still need to be investigated. There may be other non-dialyzable toxins causing post-dialysis measurement errors. The negative bias associated with post-dialysis serum creatinine indicates the above mentioned possibility.

## 5. Conclusions

The serum creatinine of uremic patients can be significantly underestimated when using conventional methods. PFOS, a type of PFC, as well the type of dialysis membrane being used, can be influencing factors.

## Figures and Tables

**Figure 1 membranes-12-00778-f001:**
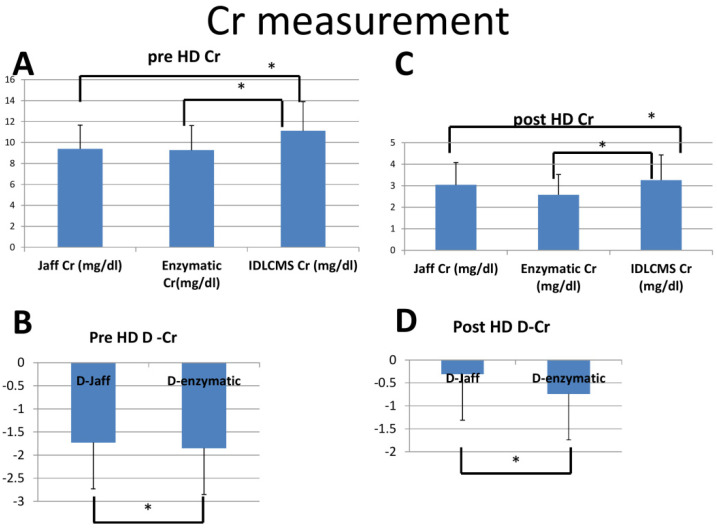
Creatinine measurement using the Jaffe and enzymatic methods as compared with the IDLCMS standard (* *p* < 0.05). (**A**) Creatinine measurement in the three methods before dialysis; (**B**) D-Jaffe and D-enzymatic before dialysis; (**C**) Creatinine measurement in the three methods after dialysis; (**D**) D-Jaffe and D-enzymatic after dialysis.

**Figure 2 membranes-12-00778-f002:**
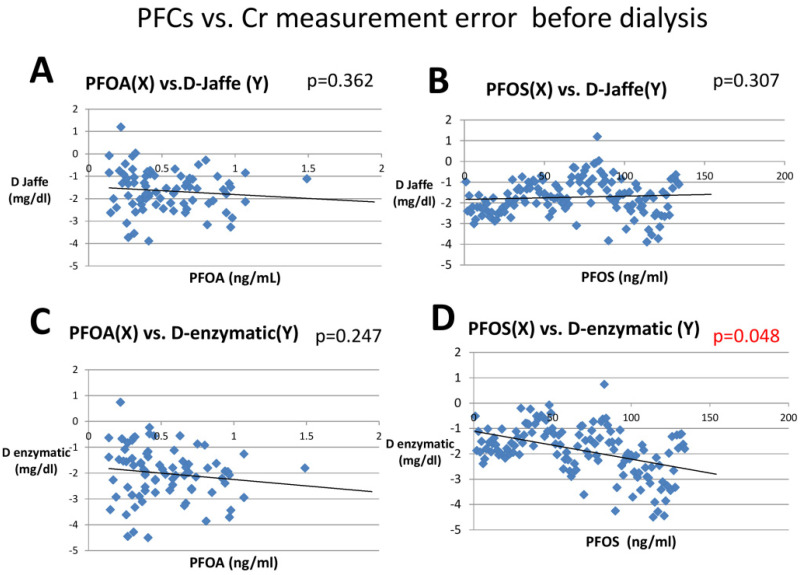
Linear regression of PFOA and PFOS with Cr difference in Jaffe and enzymatic methods (pre-dialysis). (D-Jaffe, Jaffe Cr−IDLCMS Cr; D-enzymatic, enzymatic Cr−IDLCMS Cr); (**A**) Linear regression of PFOA with D-Jaffe; (**B**) Linear regression of PFOA with D-Enzymatic; (**C**) Linear regression of PFOS with D-Jaffe; (**D**) Linear regression of PFOS with D-Enzymatic.

**Figure 3 membranes-12-00778-f003:**
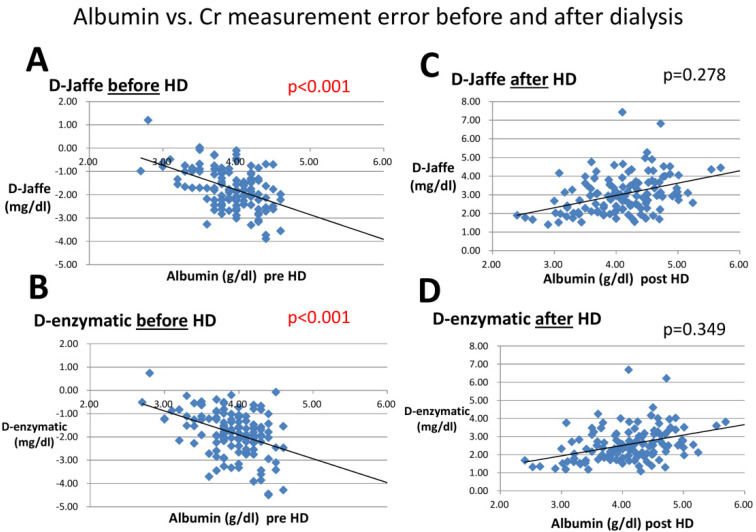
Linear regression of albumin with Cr difference using the Jaffe and enzymatic methods. (**A**) Linear regression of albumin with D-Jaffe pre-dialysis; (**B**) Linear regression of albumin with D-enzymatic pre-dialysis; (**C**) Linear regression of albumin with D-Jaffe post-dialysis; (**D**) Linear regression of albumin with D-enzymatic post-dialysis.

**Table 1 membranes-12-00778-t001:** Serum creatinine values of three measuring methods and D-Jaffe and D-enzymatic pre- and post-dialysis.

Serum Cr *n* = 124	Pre-HD Mean	±SD	*t*-Test (*p*)	Post-HD Mean	±SD	*t*-Test (*p*)
Jaffe Cr (mg/dL)	9.40	2.26	<0.001 *	3.04	1.04	<0.001 *
Enzymatic Cr(mg/dL)	9.27	2.34	<0.001 *	2.58	0.94	<0.001 *
vs. IDLCMS Cr (mg/dL) as standard	11.12	2.79		3.26	1.17	
D-Jaffe = Jaffe Cr − IDLCMS Cr	−1.73	0.86	0.023 *	−0.31	0.55	<0.001 *
D-enzymatic = enzymatic Cr − IDLCMS Cr	−1.85	0.97		−0.74	0.47	

HD, hemodialysis; SD, standard deviation; Cr, creatinine; D-Jaffe, Jaffe Cr–IDLCMS Cr; D-enzymatic, enzymatic Cr−IDLCMS Cr; IDLCMS, isotope dilution–liquid chromatography–mass spectrometry. * *p* < 0.05.

**Table 2 membranes-12-00778-t002:** Demographic, treatment, hemogram, and biochemical data with linear regression to D-Jaffe and D-enzymatic pre-hemodialysis.

N = 124	*n* (%)		D-Jaffe		D-Enzymatic	
			Mean ± SD	*p*	Mean ± SD	*p*
Male	72 (58.0%)		−1.79 ± 0.92	0.185	−1.92 ± 1.06	0.172
Female	52 (42.0%)		−1.59 ± 0.72		−1.70 ± 0.77	
DM	51 (41.1%)		−1.59 ± 0.81	0.126	−1.73 ± 0.90	0.252
Non-DM	73 (58.9%)		−1.82 ± 0.88		−1.93 ± 1.01	
			D-Jaffe		D-enzymatic	
Linear regression	Mean	±SD	(β)	(*p*)	(β)	(*p*)
Dialysis duration (month)	59.75	67.75	0.000	0.924	0.000	0.913
Dialysis frequency (/week)	2.98	0.12	0.423	0.483	0.679	0.702
Dialysis time (hours)	4.023	0.37	−0.244	0.223	−0.347	0.124
Age (year) *	59.75	14.79	0.018	0.001 *	0.017	0.002 *
Jaffe Cr (mg/dL) *	9.40	2.26	−0.230	0.000 *	−0.186	0.000 *
Enzymatic Cr (mg/dL) *	9.27	2.34	−0.192	0.000 *	−0.120	0.001 *
IDLCMS Cr (mg/dL) *	11.12	2.79	−0.220	0.000 *	−0.206	0.000 *
Ccr (mL/min) *	6.03	2.00	0.197	0.000 *	0.166	0.000 *
WBC (×1000/μL)	6.83	2.46	0.019	0.527	0.014	0.686
RBC (×10^6^/μL)	3.36	0.50	−0.242	0.099	−0.285	0.084
MCV (fl)	91.17	7.23	0.002	0.856	−0.009	0.429
Hb (g/dL) *	9.89	1.20	−0.135	0.027 *	−0.246	0.000 *
Platelet (×1000/μL)	195.71	68.31	−0.001	0.529	−0.001	0.391
Cholesterol (mg/dL) *	154.70	35.57	−0.005	0.011 *	−0.006	0.010
Glucose (mg/dL)	136.81	56.86	0.001	0.280	0.002	0.186
Total protein(gm/dL)	6.94	3.98	0.013	0.490	0.007	0.744
Albumin (gm/dL) *	3.92	0.37	−1.058	0.000 *	−0.1026	0.000 *
Globulin	2.95	4.02	0.009	0.613	0.006	0.799
AST (IU/L)	22.70	10.45	0.010	0.152	0.017	0.029 *
ALT (IU/L)	18.77	10.88	−0.001	0.891	−0.002	0.838
Alk-P (IU/L)	93.72	83.06	0.001	0.100	0.001	0.409
Total Bilirubin (mg/dL)	0.54	0.15	−0.034	0.947	0.548	0.340
Na (mEq/L) *	138.92	2.73	−0.060	0.027 *	−0.040	0.200
K (mEq/L) *	4.56	0.66	−0.232	0.041 *	−0.164	0.200
Cl (mEq/L)	98.83	5.62	−0.020	0.144	−0.016	0.297
Ca (mg/dL) *	9.27	0.86	−0.205	0.018 *	−0.287	0.003 *
P (mg/dL) *	4.69	1.33	−0.138	0.013 *	−0.059	0.352
BUN (mg/dL)	61.69	17.09	−0.010	0.021	−0.011	0.020 *
Fe (ug/dL)	59.39	22.15	−0.006	0.067	−0.010	0.007 *
TIBC (ug/dL) *	248.38	48.10	−0.003	0.029 *	0.000	0.808
Ferritin(ng/mL)	488.95	422.60	0.000	0.603	0.000	0.059
iPTH (pg/mL)	118.40	(44.17, 239.17)	0.000	0.434	8.91 × 10^−5^	0.763
Al (ng/mL)	15.16	9.85	−0.019	0.229	−0.021	0.193
PFOA (ng/mL)	0.53	0.27	−0.346	0.362	−0.494	0.247
PFOS (ng/mL) *	5.50	(1.17,24.7)	−0.002	0.307	−0.005	0.048 *

D-Jaffe, Jaffe Cr−IDLCMS Cr; D-enzymatic, enzymatic Cr−IDLCMS Cr; SD, standard deviation; DM, diabetes mellitus; IDLCMS, isotope dilution-liquid chromatography-mass spectrometry; Ccr, creatinine clearance; WBC, white blood cell; RBC, red blood cell; Hb, hemoglobin; MCV, mean corpuscular volume; AST, aspartate aminotransferase; ALT, alanine transaminase; Alk-P, alkaline phosphatase; Na, sodium; K, potassium; Cl, chloride; Ca, calcium; P, phosphorus; BUN, blood urea nitrogen; Fe, iron; TIBC, total iron-binding capacity; iPTH, parathyroid hormone; Al, aluminum; PFOA, perfluorooctanoic acid; PFOS, perfluorooctanesulfonate. Pre-dialysis PFOS and PTH variables were not normally distributed and displayed as quartile. * *p* < 0.05.

**Table 3 membranes-12-00778-t003:** Multivariate linear regression model of factors associated with D-Jaffe and D-enzymatic post-dialysis.

D-Jaffe (r^2^ = 0.352, *p* < 0.001)	B Estimate	*p*	D-Enzymatic (r^2^ = 0.435, *p* < 0.001)	B Estimate	*p*
Jaffe Cr (mg/dL) *	−0.153	0.000 *	Albumin (gm/dL) *	−0.988	0.013 *
Albumin (gm/dL) *	−0.593	0.011 *	PFOS (ng/mL) *	−0.005	0.038 *
Cholesterol (mg/dL)	−0.003	0.112	Jaffe Cr (mg/dL) *	−0.106	0.043 *
P (mg/dL)	0.088	0.156	Fe (μg/dL)	−0.007	0.130
K (meq/L)	−0.087	0.410	AST (IU/L)	0.010	0.365
Na (meq/L)	−0.010	0.697	Total Ca (mg/dL)	0.123	0.383
Age	0.002	0.760	Hb (g/dL)	−0.075	0.427
Hb (g/dL)	−0.005	0.927	Cholesterol (mg/dL)	−0.001	0.634
Total Ca (mg/dL)	0.007	0.933	BUN (mg/dL)	0.003	0.660
TIBC (ug/dL)	0.000	0.950	Age	0.001	0.870
BUN (mg/dL)	0.000	0.954			

D-Jaffe, Jaffe Cr−IDLCMS Cr; D-enzymatic, enzymatic Cr−IDLCMS Cr; P, phosphorus; K, potassium; Na, sodium; Hb, hemoglobin; Ca, calcium; TIBC, total iron-binding capacity; BUN, blood urea nitrogen; PFOS, perfluorooctanesulfonate; Fe, iron; AST, aspartate aminotransferase; BUN, blood urea nitrogen. * *p* < 0.05.

**Table 4 membranes-12-00778-t004:** Biochemical markers associated with serum albumin, Cr, and perfluorochemicals (PFCs) with linear regression to D-Jaffe and D-enzymatic post-dialysis.

*n* = 124 Post-HD	Mean	±SD	D-Jaffe		D-Enzymatic	
			(β)	(*p*)	(β)	(*p*)
Albumin (gm/dL)	4.12	0.65	0.661	0.278	0.575	0.349
Total protein (gm/dL)	7.17	1.27	0.006	0.838	−0.007	0.811
Jaffe Cr (mg/dL) *	3.04	1.04	−0.096	0.000 *	−0.206	0.000 *
Enzymatic Cr (mg/dL) *	2.58	0.94	−0.146	0.000 *	−0.208	0.000 *
IDLCMS Cr (mg/dL) *	3.26	1.17	−0.126	0.002 *	−0.071	0.000 *
BUN (mg/dL) *	15.86	5.66	−0.107	0.048 *	−0.029	0.000 *
P (mg/dL) *	1.85	0.61	−0.109	0.009	−0.208	0.000 *
K (mEq/L)	3.86	1.04	0.004	0.909	−0.024	0.522
PFOA (ng/mL)	0.53	0.34	0.012	0.927	−0.151	0.202
PFOS (ng/mL)	2.00	1.13	0.000	0.993	0.022	0.552

SD, standard deviation; D-Jaffe, Jaffe Cr−IDLCMS Cr; D-enzymatic, enzymatic Cr−IDLCMS Cr; Cr, creatinine; IDLCMS, isotope dilution–liquid chromatography-–mass spectrometry; BUN, blood urea nitrogen; P, phosphorus; K, potassium; PFOA, perfluorooctanoic acid; PFOS, perfluorooctanesulfonate. * *p* < 0.05.

**Table 5 membranes-12-00778-t005:** Serum PFOS concentration in patients with different dialysis membranes pre- and post-dialysis and linear regression of PFOS to D-Jaffe and D-enzymatic.

Dialysis Membrane	PFOS Concentration (ng/mL)	Linear Regression PFOS vs. D-Jaffe	Linear Regression PFOS vs. D-Enzymatic
		β (*p*)	β (*p*)
**Pre-HD** All (*n* = 124)	5.50 (1.17,24.7)	−0.002 (0.307)	−0.005 (0.048) *
PS (*n* = 93)	2.37 (1.17, 15.47)	−0.002 (0.247)	−0.006 (0.035) *
Non-PS (*n* = 31)	23.89 (5.93,55.53)	0.103 (0.485)	−0.002 (0.687)
*t*-test	*p* = 0.026 *		
**Post-HD** All (*n* = 124)	2.00 ± 1.13	0.000 (0.993)	0.022 (0.552)
PS (*n* = 93)	2.08 ± 1.22	−0.046 (0.430)	0.008 (0.870)
Non-PS (*n* = 31) *t*-test	1.74 ± 0.69 *p* = 0.055	0.082 (0.476)	0.152 (0.180)

PFOS, perfluorooctanesulfonate; D-Jaffe, Jaffe Cr−IDLCMS Cr; D-enzymatic, enzymatic Cr−IDLCMS Cr; HD, hemodialysis; PS: polysulfone. Pre-dialysis PFOS were not normally distributed and displayed as quartile. * *p* < 0.05.

## Data Availability

Not applicable.

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
