# Peer review of "Perfluorooctanesulfonate Can Cause Negative Bias in Creatinine Measurement in Hemodialysis Patients Using Polysulfone Dialysis Membranes"

_membranes, 2022, doi:10.3390/membranes12080778_

Round 1

Reviewer 1 Report

Wen-Shieng Liu and colleagues presents the paper focusing on new insight into protein-bound uremic toxin induced negative bias in creatinine levels estimation in uremic millieu.

The described phenomena is of the great importance since CKD is truly worldwide medical matter. All the new approaches or insights into dialysis techniques improvement are so crucial for milions of patients worldwide.

I found the presented drat interesting, albeit, I have some majors that must be addressed during the peer review process. Namely:

1) Authors withdrew to far going conclusions since protein bound uremic toxins are an extremely wide and differentiated populations. Herein, the conclusions are NOT supported by the results. Please tone down the conclusions and make them more strict, namely, address them to only one proved UT. 

2) The presentation of the data should be substantially corrected. The authors present values for iPTH as a SD when the SD value is higher than the mean. It shows that it is non-Gaussian distribution that must be presented an median plus ranges or median plus 75% IV, Moreover, the authors stated that the age of the enrolled patients was between 18 and 90 y.o. It is unacceptable since the biology of CKD, especially, of the pharmacokinetics of bounding proteins (globulins, albumins, etc) mainly depends on the age. I strongly reccomend to exclude these outfitters or divide the whole population into two groups based on their age. Right now, the presented results might be highly biased due to age discrepancies.

3) The used English language is really hard to follow and must go under the native speaker's clearance.

4) The introduction section does not provide enough information for the Readers outside the scope of the discussed area. It should be expanded.

5) Materials and Methods section is very laconic and does not allow to reproduce used LC-MS technique in another laboratory settings.

6) The whole draft should be written in the same paragraph styling and strictly follow the journal guidelines.

7) Data Availability Statement should be filled up since there is raw data underlying presented results.

Author Response

To Dear Review1

Wen-Sheng Liu and colleagues present the paper focusing on new insight into protein-bound uremic toxin induced negative bias in creatinine levels estimation in uremic milieu.

The described phenomena are of the great importance since CKD is truly worldwide medical matter. All the new approaches or insights into dialysis techniques improvement are so crucial for millions of patients worldwide.

I found the presented drat interesting, albeit, I have some majors that must be addressed during the peer review process. Namely:

1) Authors withdrew to far going conclusions since protein bound uremic toxins are an extremely wide and differentiated populations. Herein, the conclusions are NOT supported by the results. Please tone down the conclusions and make them stricter, namely, address them to only one proved UT.

Thank you for the reminder. The title is changed to “Polysulfone-based dialysis membrane proved one protein-bound toxin can cause negative bias in creatinine measurement in uremic patients”

We also mentioned as the study limitations in the last two paragraphs in the discussion section. “The limitation of the study is that we only found one toxin (PFOS) which causes a negative bias and is bound with albumin.” “More toxins and membrane materials still needed to be investigated.”

We revised the conclusion as well. “PFOS, a type of PFC, as well the type of dialysis membrane being used, can be influencing factor.”

2) The presentation of the data should be substantially corrected. The authors present values for iPTH as a SD when the SD value is higher than the mean. It shows that it is non-Gaussian distribution that must be presented an median plus ranges or median plus 75% IV.

Moreover, the authors stated that the age of the enrolled patients was between 18 and 90 y.o. It is unacceptable since the biology of CKD, especially, of the pharmacokinetics of bounding proteins (globulins, albumins, etc) mainly depends on the age. I strongly recommend to exclude these outfitters or divide the whole population into two groups based on their age. Right now, the presented results might be highly biased due to age discrepancies.

Thank you for the reminder. PTH was presented as median and interquartile range because the values were not normally distributed. We revised it accordingly and there is an explanation in the footnote of Table 2.

As for age factor, the age of uremic patients is 59.75±14.79, which is normally distributed. Furthermore, the enrollment for uremic patients does not guarantee they are stable under dialysis for more than 3 months. Older patients are more prone to diseases and were likely to be excluded. (In the second paragraph of the methods section, we indicated that “Those who had blood loss with transfusion or admission were also excluded.”)

We appreciated and agreed with your suggestion that age may affect serum albumin. Older age was correlated with lower albumin and therefore has lower negative bias. This was shown in the Table 2 (p=0.001 for D-Jaffe and p=0.002 for D-enzymatic). However, age lost its influence in the multivariate linear regression (Table 3). Therefore, the main factor affecting creatinine is albumin rather than age.

3) The used English language is really hard to follow and must go under the native speaker's clearance.

Thank you for the reminder. We have the manuscript checked with a native speaker and have revised the whole manuscript thoroughly.

4) The introduction section does not provide enough information for the Readers outside the scope of the discussed area. It should be expanded.

Thank you for the reminder. We will revise it with more information and explain positive and negative bias for easier reading (Positive bias means that the measured value is higher than the real value and negative bias means that the measured value is lower than the real value).

5) Materials and Methods section is very laconic and does not allow to reproduce used LC-MS technique in another laboratory settings.

We have provided the procedures for measuring serum creatinine and PFCs with IDLCMS in the supplementary material.

6) The whole draft should be written in the same paragraph styling and strictly follow the journal guidelines.

Thank you for the reminder. We will check the format with the editor and revise accordingly with the journal guidelines.

7) Data Availability Statement should be filled up since there is raw data underlying presented results.

We will upload the excel file once the link is provided.

Reviewer 2 Report

To the authors

Thank you for the opportunity to review this article.The authors investigated the effect of PFCs accumulation on Jaffe method and enzymatic method in serum creatinine measurement, and showed PFOS cause negative bias in enzymatic method. This report is very interesting and well-written manuscript. However, problems as described below should be resolved for the manuscript to be accepted for publication in Membranes.

I have only two comments.

<Specific comments>

1. Are the D-jaff and D-enzymatic different between HD and HDF? For example, from this study, serum Alb affects both D-jaff and D-enzymatic; however, the amount of albumin leaked by dialysis differs between HD and HDF. Furthermore, is there a difference in the removal of PFCs between HD and HDF?

2. In Table 4, why the number of total patients is 133? Isn't the number of patients 124?

Author Response

Thank you for the opportunity to review this article.The authors investigated the effect of PFCs accumulation on Jaffe method and enzymatic method in serum creatinine measurement, and showed PFOS cause negative bias in enzymatic method. This report is very interesting and well-written manuscript. However, problems as described below should be resolved for the manuscript to be accepted for publication in Membranes.

I have only two comments.

<Specific comments>

  1. Are the D-jaff and D-enzymatic different between HD and HDF? For example, from this study, serum Alb affects both D-jaff and D-enzymatic; however, the amount of albumin leaked by dialysis differs between HD and HDF. Furthermore, is there a difference in the removal of PFCs between HD and HDF?

Our study is focused on regular hemodialysis. We have not tested on HDF yet.

We agreed with the possibility that D-Jaffe and D-enzymatic may be different between HD (hemodialysis) and HDF (hemodiafiltration) if the albumin level is changed.

We are currently testing the difference of PFC removal on different dialysis methods, such as HDF and other dialysis modalities However, the population number may not be the adequate enough as of now to produce statistically significant results due to dialysis being an invasive procedure with possible complications of bleeding and infection due to puncture.

We only recruited eligible patients who were indicated for such treatment (for example, patients with hyperlipidemia who underwent double filtration to remove serum lipid). We hope to share these result in the near future with you.

  1. In Table 4, why the number of total patients is 133? Isn't the number of patients 124?

Thank you for your reminder. We have corrected the error accordingly to 124 patients as the total number of the study population.

Round 2

Reviewer 1 Report

The Authors in the correct way addressed all my majors and most of the minors.

The raised issues have been fixed and the paper gained much more scientific soundness. 

At this stage, I would like to recommend the paper for the acceptance as it is.